# SCALABLE MODULAR NETWORK: A FRAMEWORK FOR ADAPTIVE LEARNING VIA AGREEMENT ROUTING

**Minyang Hu**[1,2]**, Hong Chang**[1,2]**, Bingpeng Ma**[2]**, Shiguang Shan**[1,2]**, Xilin Chen**[1,2]
[1] Institute of Computing Technology, Chinese Academy of Sciences
[2]University of Chinese Academy of Sciences
`minyang.hu@vipl.ict.ac.cn, bpma@ucas.ac.cn`
`{changhong, sgshan, xlchen}@ict.ac.cn`

## ABSTRACT

In this paper, we propose a novel modular network framework, called Scalable Modular Network (SMN), which enables adaptive learning capability and supports integration of new modules after pre-training for better adaptation. This adaptive capability comes from a novel design of router within SMN, named agreement router, which selects and composes different specialist modules through an iterative message passing process. The agreement router iteratively computes the agreements among a set of input and outputs of all modules to allocate inputs to specific module. During the iterative routing, messages of modules are passed to each other, which improves the module selection process with consideration of both local interactions (between a single module and input) and global interactions involving multiple other modules. To validate our contributions, we conduct experiments on two problems: a toy min-max game and few-shot image classification task. Our experimental results demonstrate that SMN can generalize to new distributions and exhibit sample-efficient adaptation to new tasks. Furthermore, SMN can achieve a better adaptation capability when new modules are introduced after pre-training. Our code is available at https://github.com/hu-my/ScalableModularNetwork.

## 1 INTRODUCTION

Inferring new knowledge from existing information is a hallmark of human intelligence. For instance, an individual who understands two concepts of "wooden table" and "iron chopsticks" can immediately understand a new concept of "wooden chopsticks", even without having seen any examples of it (Atzmon et al., 2020). This learning efficiency comes from the cognitive ability in human brain that decomposes knowledge into abstract sub-concepts and then recomposes these acquired sub-concepts to understand a new concept (Sternberg, 2011; Meunier et al., 2009). Through this *modular cognition process*, humans adapt well to the change of environment and excel at dealing with new tasks (Sporns & Betzel, 2016).

To realize such learning ability in machines, *modular neural networks* are introduced as a promising solution. A modular neural network involves several specialist modules, which serve as computation units to process specific sub-concepts. A natural question arises: *How to compose the specialist modules for different inputs?* Prior works utilize metadata to control the information flow to modules and make discrete routing decision before training. For example, some works (Andreas et al., 2016; Hu et al., 2017a;b; Buch et al., 2021) route different specialist modules for visual question answer with linguistic syntax tree. Besides, some vision-language methods (Radford et al., 2021; Tsimpoukelli et al., 2021; Alayrac et al., 2022; Najdenkoska et al., 2023) design deterministic and fixed routing based on modality information, thus the visual and linguistic modules are responsible for different modality streams. Although these works can decompose knowledge into modular parts and then recompose, they require extra metadata for routing, which does not always exist.

Learning the *routing function* from training data with some inductive biases is more promising for modular neural networks. One popular routing strategy is routing by input, where a router selects different modules to activate conditioned on the input data (Shazeer et al., 2017; Li et al., 2022; Ma

et al., 2018; Rosenbaum et al., 2018; Kirsch et al., 2018; Chang et al., 2019). To avoid a trivial solution where a router always selects all modules or a few modules repeatedly for all inputs, these methods often integrate reinforcement learning algorithm (Rosenbaum et al., 2018; Kirsch et al., 2018; Chang et al., 2019) or a noisy top-$k$ selection mechanism (Shazeer et al., 2017; Li et al., 2022; Ma et al., 2018). Despite the promising results shown by the routing-by-input approach, these works ignore the role of module states in routing process and cannot effectively handle situations where new modules need to be inserted. To overcome this limitation, recent works (Goyal et al., 2021b; Rahaman et al., 2021; 2022) propose routing based on the pairwise interaction between each module state and input. NI (Rahaman et al., 2021) and NAC (Rahaman et al., 2022) divide module parameters into two parts: a signature vector and a computation part, where the former decides routing through a pairwise attention mechanism. RIMs (Goyal et al., 2021b) extend the top-$k$ mechanism based on all pairwise interactions between module states and model inputs. However, these studies ignore the global interactions during module selection process, which plays an important role for selecting appropriate modules to differentiate similar samples.

In this work, we introduce a novel modular network framework, called **Scalable Modular Network (SMN)**, to remedy the above limitations with *adaptive learning* capability. To this end, we design *agreement router* for dynamic module selection with consideration of both local interaction (between input and a single module) and global interactions (among all modules). More specifically, the agreement router selects and composes different specialist modules through an iterative message passing process, during which top-down feedback is iteratively obtained thus both local and global interactions are counted. In this way, the agreement router enables adaptive learning capability, i.e. adapting to specific input and refining the module composition by iterative explorations.

Moreover, SMN supports integration of new modules after model training for better adaptation. When new modules are introduced, the agreement router dynamically reassigns coefficients of all modules based on the inputs and the functionality of existing modules (i.e. their corresponding outputs), enabling a *scalable* routing process to enhance the adaptation ability of the modular network. Overall, leveraging the adaptive learning capability, SMN can effectively select and compose distinct modules to represent new samples within the representation space, thereby facilitating the out-of-distribution generalization and task adaptation.

**Our primary contributions are as follows.** **(a)** We introduce Scalable Modular Network, a novel modular framework that enables adaptive learning capability and supports integration of new modules for better adaptation. **(b)** We propose agreement router for module composition that considers both local and global module interactions with iterative message passing. **(c)** We evaluate the proposed SMN architecture and agreement router in two learning problems that requires generalization and fast adaptation capabilities. We show that SMN is capable of sample-efficient adaptation and scalable with new modules. In addition, SMN can effectively distinguish similar sub-concepts by utilizing global information.

## 2 RELATED WORKS

**Modular Neural Networks.** A modular neural network comprises multiple specialist modules that exhibit properties of reusability and combinability. These properties enable modular architectures to possess good compositional generalization ability. To incorporate networks with modular inductive bias, previous research efforts (Andreas et al., 2016; Hu et al., 2017a; Rahaman et al., 2021; 2022; Goyal et al., 2021b) have explored various approaches. A direct solution is to employ the metadata (Andreas et al., 2016; Hu et al., 2017a; Buch et al., 2021), such as natural language or modality information, to decide the routing before learning modules separately. Alternatively, other approach (Goyal et al., 2021a;b; Rahaman et al., 2021; 2022) explores neuro-symbolic architectures and utilize distinct strategies to simultaneously learn modules and routing functions without relying on extra metadata. In our work, the proposed SMN aligns with the latter approach, but with enhanced adaptation and scalable ability.

**Module Composition.** An important problem in modular network is how to select and compose specialist modules for different input samples, which is related to the compositional generalization of modular neural networks (Bahdanau et al., 2019; Rosenbaum et al., 2019). Various routers have been designed for this purpose. For instance, MoE (Shazeer et al., 2017; Li et al., 2022; Ma et al.,

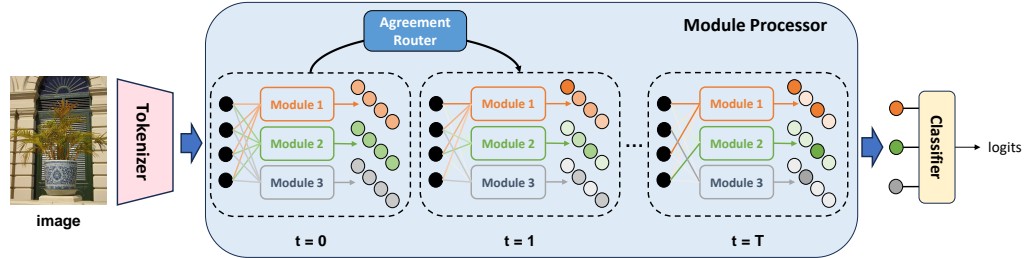

Figure 1: A sketch of the proposed SMN. The SMN contains three components: (a) a tokenizer that transforms an input image into local features; (b) a module processor containing specialist modules and an agreement router; (c) a classifier that aggregates module outputs for final classification.

2018) employs a gating network to select specialist modules based on module input. Top-$k$ selection mechanism is incorporated to restrict the number of modules to be activated and updated for each sample. RIMs (Goyal et al., 2021b) extend the top-$k$ mechanism to select RNN-based modules according to the pairwise interactions between each module state and input. Besides, NI (Rahaman et al., 2021) divides the module parameters into a signature vector and a computation part. The signature vector specifies the most relevant input vectors for the corresponding module by a pairwise kernel function. NAC (Rahaman et al., 2022) inherits the same router design but imposes a graph structural prior on the learning process. In contrast to RIMs and NAC, SMN selects modules based on agreement between inputs and outputs of all modules. This routing strategy incorporates global interactions during module selection process, and is compatible with arbitrary module designs.

**Fast Adaptation on New Tasks.** How to learn new tasks with very few labeled samples by leveraging experience from related training tasks is a central question in few-shot learning (FSL) area (Hu et al., 2023). Previous FSL approaches based on meta-learning framework can be categorized into two categories. The metric-based approach (Vinyals et al., 2016; Snell et al., 2017; Sung et al., 2018; Hou et al., 2019; Allen et al., 2019) aims to learn a cross-task embedding function and predict the query labels based on the learned distances. The optimization-based approach focuses on learning some optimization states, like model initialization (Finn et al., 2017; Sun et al., 2019) or step sizes (Antoniou et al., 2019; Li et al., 2017), to rapidly update models with very few labeled samples. Recent FSL works (Dong et al., 2022; Hiller et al., 2022; Lin et al., 2023) have shown that standard transfer learning procedure, which involves initial pre-training with self-supervised loss and subsequent fine-tuning, achieves state-of-the-art performance on various benchmarks. In this work, we demonstrate that the transfer-learning based approach can be combined with SMN framework and results in a further performance improvement due to the modular inductive bias.

## 3 SCALABLE MODULAR NETWORK

In this work, we present a novel modular framework, **Scalable Modular Network (SMN)**, that enables adaptive learning capability and supports integration of new modules after pre-training. In Section 3.1, we will introduce three components that constitute the SMN framework. Next, in Sections 3.2 and 3.3, we will describe how to route different specialist modules based on input-output agreements and explain how the agreement routing algorithm handles scenarios where new modules are added to the pre-trained network. Section 3.4 will focus on the training process for both modules and agreement router within the SMN framework. Finally, in Section 3.5, we will discuss the relationship between agreement routing and other works.

### 3.1 OVERVIEW OF SMN

The SMN consists of three components: (**a**) a tokenizer that transforms an input image to a set of local features, (**b**) a module processor which contains multiple modules and an agreement router that composes a subset of specialist modules for specific input, (**c**) a classifier that aggregates the outputs from the module processor to make final predictions. A sketch of SMN is presented in Fig. 1.

**Tokenizer.** The tokenizer is any component that converts the input image to a set of representation vectors. It can be the feature extractor of a ConvNet without global pooling, or a transformer encoder. We try both two choices in the experiments.

**Module Processor.** The module processor is composed of an agreement router and multiple specialist modules. The agreement router dynamically selects a subset of specialist modules to be activated through an iterative process. After that, each module computes using its own parameters with the acquired input vectors and then outputs a single vector. These output vectors of modules will be forwarded to the classifier for further processing.

**Classifier.** The classifier is responsible for receiving the output vectors from the module processor and subsequently mapping them to the label space. It is feasible to aggregate these output vectors from the module processor through average or concatenation operation to acquire a single vector for final classification.

### 3.2 ITERATIVE ROUTING BY AGREEMENT

The kernel part of SMN is the agreement router, which selects different specialist modules through iterative agreement routing algorithm. Assume that the tokenizer outputs $N$ vectors $\{s_1, ..., s_N\}$, and the module processor contains $M$ modules $\{f_1, ..., f_M\}$. The agreement routing algorithm maps from a set of $N$ input vectors to the output vectors $\{v_1, ..., v_M\}$ of $M$ modules, as $v_j = f_j(\{s_i\}_{i=1}^N)$. The routing process iterates with two steps: computing input-output agreements and updating output vectors. First, the agreement router transforms each input vector through a learnable matrix and computes the agreement between transformed vector and each module output. Then, the router selects different input vectors for $M$ modules and refines their outputs.

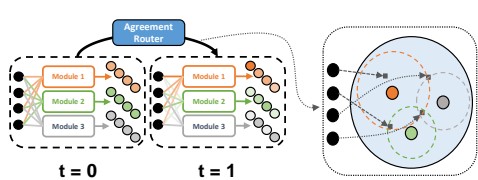

Figure 2: Agreement routing with two steps. Left: selecting inputs for distinct modules and refining their outputs. Right: computing the agreements between transformed input vectors (dark squares) and module outputs (three colored circles).

An illustration of agreement routing is shown in Fig. 2. Each module output can be regarded as a cluster and the input vectors can be viewed as samples to be grouped. More formally, for a set of input vectors $\{s_1, ..., s_N\}$, we first apply all available modules to each input vector $s_i$ to get the processed vectors $\{u_{ij} = f_j(s_i)\}_{j=1}^M$. We then calculate the average of the corresponding processed vectors $v_j^{(0)} = \frac{1}{N}\sum_{i=1}^N u_{ij}$ as the initial output of each module $f_j$. At the $t$-th iteration, we calculate the agreement score $\alpha_{ij}^{(t)}$ between each input vector $s_i$ and module output $v_j^{(t-1)}$ by cosine similarity:

$$\alpha_{ij}^{(t)} = \begin{cases} 0, & t = 0 \\ \dfrac{v_j^{(t-1)} * \boldsymbol{W}_a s_i}{\|v_j^{(t-1)}\|\|\boldsymbol{W}_a s_i\|} + \alpha_{ij}^{(t-1)}, & t \neq 0 \end{cases}, \tag{1}$$

where $\boldsymbol{W}_a$ is a learnable matrix that maps $s_i$ into the module output space, and $*$ denotes the dot-product operation. We normalize the agreement score $\alpha_{ij}^{(t)}$ with a softmax function to get the agreement coefficient $c_{ij}^{(t)}$ which is then used to update the output vector of the $j$-th module:

$$c_{ij}^{(t)} = \frac{\exp(\alpha_{ij}^{(t)})}{\sum_{j'=1}^M \exp(\alpha_{ij'}^{(t)})}, \qquad v_j^{(t)} = \sum_{i=1}^N c_{ij}^{(t)} u_{ij}. \tag{2}$$

Through the softmax function, each module competes with others in every iteration to decide to what extent the input vector can be used to acquire its output vector. Thus, the agreement coefficients determine how the modules are activated, conditioned on the input-output agreements. The agreement routing process will alternate between Eq. 1 and Eq. 2 until the number of iteration reaches a threshold or the agreement coefficients no longer change. The process of agreement routing is summarized in Algorithm 1.

---

**Algorithm 1** Agreement Routing.

---
**Input**: $\{u_{1j}, ..., u_{Nj}\}_{j=1}^{M}, \{s_1, ..., s_N\}$
**Parameter:** $\boldsymbol{W}_a$

1: Initialize $v_j^{(0)} \leftarrow \frac{1}{N} \sum_{i=1}^{N} u_{ij} \in R^{D_d}$, $\alpha_{ij}^{(0)} \leftarrow 0$.
2: **while** $t = 1 \rightarrow T$ **do**
3:    for all input vector $s_i$ and output vector $v_j$: $\alpha_{ij}^{(t)} \leftarrow \alpha_{ij}^{(t-1)} + \frac{v_j^{(t-1)} * \boldsymbol{W}_a s_i}{\|v_j^{(t-1)}\|\|\boldsymbol{W}_a s_i\|}$
4:    for all input vector $s_i$ and output vector $v_j$: $c_{ij}^{(t)} \leftarrow \frac{\exp(\alpha_{ij}^{(t)})}{\sum_{j'} \exp(\alpha_{ij'}^{(t)})}$
5:    for all output vector $v_j$: $v_j^{(t)} \leftarrow \sum_i c_{ij}^{(t)} u_{ij}$
6: **end while**
7: **return** $\{v_1^T, ..., v_M^T\}$

---

**Sparse Module Selection.** As the agreement routing algorithm does not explicitly impose constraints on the number of activated modules, a natural question arises that how does it enable a sparse module selection so as to avoid trivial solutions? We explore this question theoretically and discover that, as the iterative computation proceeds, the agreement routing naturally tends to produce a sparse module selection.

**Theorem 1.** *Let $\{s_1, ..., s_N\}$ be N input vectors and $\{u_{ij} = f_j(s_i)\}_{j=1}^{M}$ a set of output vectors of M modules $\{f_1, ..., f_M\}$. As for the agreement routing alternating between Eq. 1 and Eq. 2, if any two input vectors are not identical, the coefficient $c_{ij} \rightarrow 0$ or 1 as the number of iterations $T \rightarrow \infty$.*

*Proof.* Please refer to Appendix A.1. □

Theorem 1 shows that as the iteration number $T \rightarrow \infty$, each input vector will ultimately be assigned to a single specialist module based on the sign of agreements. This implies that there will only exist $N$ connections between input vectors and specialist modules, avoiding the risk of trivial solutions.

### 3.3 ROUTING WITH NEW MODULES

After training, the agreement router within a SMN should dynamically select and compose distinct specialist modules for different input samples. However, it is possible that the downstream tasks introduce unseen visual primitives or have new feature distributions. Under such circumstances, the trained modules may be inadequate in adapting to the environment changes. Therefore, it becomes necessary to incorporate new modules that are not only complementary with existing modules but also adaptable to the environment changes.

The agreement routing algorithm can naturally handle the situation where new modules are added into a trained model. Assume that a SMN with $M$ modules $\{f_1, ..., f_M\}$ and an agreement router has been trained, and $D$ new modules $\{f_1', ..., f_D'\}$ are being added into this model. We utilize the trained agreement router to route the existing and new introduced modules simultaneously. For each new module $f_j'$, we calculate the agreement score $\alpha_{ij'}^{(t)}$ and utilize a softmax function across all $M + D$ modules to determine the agreement coefficient $c_{ij'}^{(t)}$ and refine its output vector $v_{j'}^{(t)}$ as:

$$c_{ij'}^{(t)} = \frac{\exp(\alpha_{ij'}^{(t)})}{\sum_{j'=1}^{D} \exp(\alpha_{ij'}^{(t)}) + \sum_{j=1}^{M} \exp(\alpha_{ij}^{(t)})}, \qquad v_{j'}^{(t)} = \sum_{i=1}^{N} c_{ij'}^{(t)} f_j'(s_i). \qquad (3)$$

Through the softmax function, the $D$ new modules are introduced into the trained model, where they compete with the existing $M$ modules. This competitive process serves to allocate the new modules' attention to distinct inputs, achieving complementary effects with the existing modules.

### 3.4 LEARNING MODULES AND AGREEMENT ROUTER

We jointly train specialist modules and an agreement router using a combination of two losses: classification loss $\mathcal{L}_{cls}$ and importance loss $\mathcal{L}_{imp}$. The classification loss is computed on the classifier

component with aggregated outputs from the modules $\{v_j^{(T)}\}_{j=1}^M$. This loss guides the agreement router to select appropriate modules to classify the input sample:

$$\mathcal{L}_{cls} = \sum_{x,y} \text{CE}(o, y), \tag{4}$$

where $o$ is output logits from the classifier for input sample $x$, and CE denotes the cross entropy loss. However, we have observed that only using the classification loss consistently leads the router to select the same set of a few modules for different samples, as illustrated in (Shazeer et al., 2017). To address this issue, an importance loss is introduced to balance the utilization of specialist modules based on their agreement coefficients $c_{ij}$. Following (Shazeer et al., 2017), we define the importance loss on a set of importance values $\{imp(f_j)\}_{j=1}^M$ as the square of the coefficient of variation (CV):

$$\mathcal{L}_{imp} = \text{CV}(\{imp(f_j)\}_{j=1}^M)^2, \tag{5}$$

where $imp(f_j) = \sum_{i=1}^N c_{ij}^{(T)}$ with $c_{ij}^{(T)}$ being the agreement coefficient for input $s_i$ and module $f_j$ at iteration $T$. The final objective $\mathcal{L}$ combines both two losses:

$$\mathcal{L} = \mathcal{L}_{cls} + \lambda \mathcal{L}_{imp}. \tag{6}$$

Here, $\lambda$ is a hyper-parameter that balances the contributions of the two losses.

**Fine-tuning with New Modules.** When a pre-trained SMN needs to adapt to a new task by adding new modules, we fix all existing pre-trained parameters and only fine-tune the newly added modules. This partial fine-tuning strategy reduces the required number of samples and avoids catastrophic forgetting to enable the boarder generalization. Furthermore, most of the parameters of the new modules can be shared with pre-trained modules, and only a small set of parameters have to be tuned for their unique computations. Consequently, the total number of parameters does not noticeably increase with the number of new modules, resulting in a sample-efficient adaptation process.

## 3.5 DISCUSSION

**Agreement Routing as Parse Tree Constructing.** Modular neural network aims to factorize knowledge into multiple modular sub-concepts, which can be reused and recomposed to solve out-of-distribution tasks. This idea of knowledge factorization is similar with a cognitive theory, known as "recognize by components" (Biederman, 1987), which suggests to construct a visual parse tree to represent an image. One challenge is how to dynamically assign the low-level nodes to high-level nodes in the visual parse tree. To address this issue, a clustering-like process can be applied (Sabour et al., 2017), which regards each high-level node as a cluster and low-level nodes as samples to be grouped. By iteratively computing the similarity between samples and clusters, each low-level node will be dynamically assigned to appropriate high-level nodes. From this point of view, the proposed agreement routing within SMN is equivalent to constructing a visual parse tree to represent an image. Each node in the tree corresponds to a specialist module responsible for a specific sub-concept.

**Agreement Routing and $k$-means Clustering.** Agreement routing is similar with $k$-means algorithm, a classical unsupervised learning approach which divides the training set into $k$ distinct clusters of examples that are close to each other. In contrast to $k$-means, the agreement routing algorithm is driven by both a supervised classification loss and an importance loss. This routing approach aims to adaptively select and compose different specialist modules for classification. Furthermore, in agreement routing process, the computed agreements at previous step are accumulated for the next step, which encourages a sparse module selection to mitigate the risk of trivial solutions.

## 4 EXPERIMENTS

In this section, we empirically evaluate Scalable Modular Network in various problem settings, aiming to address the following key questions: (1) Can the agreement router improve module selection process by incorporating global interactions? (2) Can SMN exhibit better generalization and adaptation capabilities? (3) Can SMN support integration of new modules after pre-training for better adaptation? We first address these questions through a toy example and subsequently validate the applicability of SMN in a more realistic scenario.

## 4.1 TOY PROBLEM: MIN-MAX GAME

We first consider a simple Min-Max Digit Game, where a model is required to find the minimum or maximum digit according to the summation of all digits in the image. Our hypothesis is if a modular model can incorporate global interactions into the module selection process, it can differentiate distinct samples even if some parts of them are very similar.

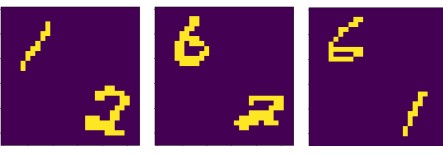

Figure 3: Visualization of synthetic images. Left and Middle: two synthetic images from training digital compositions (1,2) and (6,2). Right: a synthetic image from test digital composition (6, 1).

| Method | | Params | Test Acc |
|---|---|---|---|
| **CNN** | | 128.9K | 42.34 (2.7) |
| **Transformer** | | 125.7K | 45.76 (1.8) |
| **Top-K** | | 124.6K | 42.94 (2.1) |
| **Truncated** | | 131.9K | 43.77 (3.4) |
| **SMN** | $T = 0$ | | 44.55 (2.2) |
| | $T = 2$ | 131.9K | 47.60 (4.1) |
| | $T = 4$ | | **54.46 (5.0)** |

Table 1: Results on out-of-distribution samples.

**Task Construction.** Consider a set of $N$ digital images, denoted as $\{(x_i, d_i)\}_{i=1}^N$, where each $x_i$ represents an image sample and $d_i \in [0, 9]$ signifies its corresponding digit label. We create a synthetic image along with a corresponding label using any two elements with the following operation:

$$x = \text{concatenate}(x_i, x_j); \quad y = \begin{cases} \min(d_i, d_j), & d_i + d_j \geq 10 \\ \max(d_i, d_j), & d_i + d_j < 10 \end{cases} \quad (7)$$

In the above operation, we combine two digital images and assign the resulting synthetic image a label equal to the minimum or maximum value of the two digits according to their summation. We select 55 different compositions of digits to construct a Min-Max Digit task, and split 40 compositions for training and another 15 compositions for test, respectively. To evaluate the out-of-distribution generalization ability, we constrain the synthetic image for test to only include new compositions of two digits, but each individual digit must have been encountered during the training phase. See some synthetic examples in Fig. 3.

**Comparative Methods.** We compare our SMN with two non-modular network architectures: CNN and Transformers. Furthermore, we replace the agreement router in SMN with two module selection methods (Shazeer et al., 2017; Rahaman et al., 2021): noisy top-$k$ selection mechanism (Top-K) and truncated kernel (Truncated). We set the number of modules $M$ to 2 for SMN, with the expectation that these two specialist modules will learn to find the minimum or maximum digit in a digital composition. For a fair comparison, we set the same number of modules for the two modular network variants. See more experimental details in the Appendix C.

**Out-of-distribution Generalization.** A modular model should excel at composing specialist modules for out-of-distribution generalization. Thus, if the agreement router improves module selection process, SMN is expected to exhibit a better generalization performance on the new digital compositions. Tab. 1 shows the results of different models. First of all, when the the iteration number $T$ is 0, the performance of SMN is similar with non-modular models, CNN and Transformers. However, SMN can significantly outperform them with a large $T$. This phenomenon shows the importance of iterative computation in agreement routing. Secondly, in comparison with two modular variants (Top-K and Truncated), our SMN generalizes better on new compositions than them. We attribute this to the global interactions incorporated in agreement router, which is absent for the two methods. Futhermore, we visualize the activation map of two modules within SMN. In Fig. 4, we observe that the module $f_1$ focuses on digits 6 and 4 in the first example while becomes non-activated in the second example, even though the digit 6 appears in both examples. This verifies our hypothesis that incorporating global information helps the specialist modules to separate unseen similar samples, thus improving the generalization capability of the model.

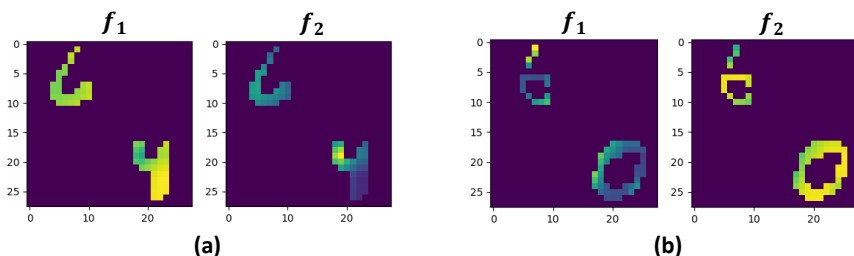

**(a)**                                    **(b)**

Figure 4: Activation maps of two modules within the SMN ($T = 4$). **(a)** example from composition (6,4) and label $y = \min(6, 4)$. **(b)** example from composition (6,0) and label $y = \max(6, 0)$.

| Method | | Test Accuracy | | | | | |
|---|---|---|---|---|---|---|---|
| | | $k = 15$ | $k = 30$ | $k = 45$ | $k = 60$ | $k = 75$ | $k = 90$ |
| **CNN** | | 82.9 (2.4) | 84.8 (2.0) | 85.9 (2.6) | 87.8 (1.6) | 88.6 (1.3) | 89.5 (1.3) |
| **Transformer** | | 82.2 (2.4) | 85.0 (3.1) | 86.4 (1.6) | 87.4 (1.6) | 88.8 (1.2) | 88.6 (0.9) |
| **Top-K** | | 83.2 (3.9) | 85.3 (2.3) | 86.5 (1.6) | 87.4 (1.1) | 88.1 (1.0) | 88.2 (0.6) |
| **Truncated** | | 82.7 (3.3) | 85.9 (1.9) | 86.9 (1.3) | 87.7 (0.9) | 87.8 (1.2) | 88.0 (0.7) |
| **SMN** | $T = 0$ | 84.4 (2.7) | 86.9 (2.2) | 88.2 (1.7) | 89.7 (1.0) | 89.6 (1.3) | 90.3 (1.3) |
| | $T = 2$ | 86.4 (3.1) | **89.0 (2.2)** | **90.4 (1.5)** | **91.7 (0.8)** | **91.8 (0.9)** | 91.8 (1.0) |
| | $T = 4$ | **87.8 (1.6)** | 87.3 (1.6) | 89.0 (1.6) | 90.6 (1.7) | 91.3 (1.0) | **92.3 (1.0)** |

Table 2: Results of various models for few-shot task adaptation. We run each experiment 10 times with different random seeds and report the average results with standard deviation.

**Few-shot Adaptation.** We further explore the task adaptation capability of SMN on a transfer learning scenario. We design a new task based on parity code of digital compositions. In this task, the label of a digital composition depends on the count of value 1 in the binary encoding of minimum or maximum digit. Specifically, if the count is odd, the corresponding label is 1, otherwise it equals to 0. We train various models on the Min-Max Digit task with sufficient training samples, and then fine-tune them on the Parity Code task with $k$ labeled samples. Tab. 2 shows that SMN is better than all other models with different values of $k$. And, this performance improvement becomes more significant with the increase of iteration number $T$. This comparison shows the importance of agreement routing on task adaptation capability for SMN. Besides, we verify that SMN supports integration of new modules after pre-training for better adaptation in Fig. 5. We observe that the performance can be further improved by incorporating a small number of new modules into a pre-trained SMN. However, as we increase the number of added modules, the improvement diminishes. We attribute this to the overfitting, as the incorporation of new modules introduces more parameters.

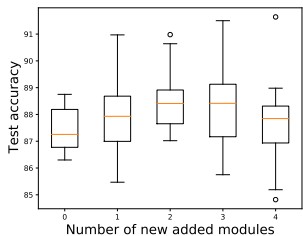

Figure 5: Test accuracy for SMN ($T = 4$) in terms of the number of new added modules.

## 4.2 FEW-SHOT IMAGE CLASSIFICATION TASK

We have shown that SMN exhibits better generalization and adaptation capabilities on new compositions. While these improvement have been striking in synthetic dataset, it raises a question: Can SMN still perform well on a more realistic scenario? To answer the question, we consider the Few-Shot Image Classification (FSIC) task, which aims to learn novel categories with very few labeled samples by exploiting experience from related training categories. Our intuition lies in the attribute compositionality hypothesis: each category can be represented as a composition of attributes, which are reusable in a huge assortment of meaningful compositions (Hu et al., 2023). We conjecture that when applied to FSIC task, SMN should have a better task adaptation capability as its modular inductive bias leads to improved generalization on previous unseen compositions.

| Method | *mini*ImageNet | CUB |
|---|---|---|
| RelationNet | 49.31 ± 0.85 | 62.34 ± 0.94 |
| MAML | 46.47 ± 0.82 | 54.73 ± 0.97 |
| ProtoNet | 44.42 ± 0.84 | 50.46 ± 0.88 |
| Baseline++ | 48.24 ± 0.75 | 60.53 ± 0.83 |
| SMN | **54.35 ± 0.77** | **67.33 ± 0.90** |

Table 3: Results on 5-way 1-shot classification with Conv4-64 backbone.

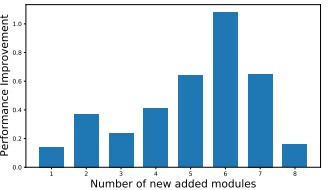

Figure 6: 1-shot improvement with different number of added modules on *mini*ImageNet.

| Method | Backbone | *mini*ImageNet | | CIFAR-FS | |
|---|---|---|---|---|---|
| | | 5-way 1-shot | 5-way 5-shot | 5-way 1-shot | 5-way 5-shot |
| SUN | ViT | 67.80 ± 0.45 | 83.25 ± 0.30 | 78.37 ± 0.46 | 88.84 ± 0.32 |
| FewTURE | Swin-Tiny | 72.40 ± 0.78 | 86.38 ± 0.49 | 77.76 ± 0.81 | 88.90 ± 0.59 |
| SMKD | ViT-S | 74.28 ± 0.18 | 88.82 ± 0.09 | 80.08 ± 0.18 | 90.63 ± 0.13 |
| SMN | ViT-S | **75.76 ± 0.84** | **89.28 ± 0.47** | **82.87 ± 0.79** | **91.35 ± 0.56** |

Table 4: Results on 5-way 1-shot and 5-shot classification on *mini*ImageNet and CUB.

**Datasets and Setup.** Following (Chen et al., 2019; Lin et al., 2023), we consider three popular benchmark in FSIC task: *mini*ImageNet (Vinyals et al., 2016), CIFAR-FS (Bertinetto et al., 2018) and CUB-200-2011 (CUB) (Wah et al., 2011). As previous works, we use the $N$-way $K$-shot setting to construct training and test episodes from dataset, where $K$ labeled samples are provided for each of $N$ classes. We report the average accuracy on 600 test episodes with 95% confidence intervals. See more details of datasets and experimental setup in the Appendix C.5.

**Comparative Methods.** We compare our model with four classical few-shot methods: (1) MAML (Finn et al., 2017), (2) ProtoNet (Snell et al., 2017), (3) RelationNet (Sung et al., 2018) and (4) Baseline++ (Chen et al., 2019). These methods use the Conv4-64 network as backbones. Further more, we include three SOTA few-shot methods: (1) SUN (Dong et al., 2022), (2) FewTURE (Hiller et al., 2022), (3) SMKD (Lin et al., 2023), which use the vision transformer as backbones.

**Results.** Tab. 3 and Tab. 4 show the results of various models on three datasets. Compared to other FSL method, our SMN achieves the best performance on all datasets, which shows modular architecture design do improve the fast adaptation ability. With the same transformer backbone, our SMN can achieve new state-of-the-art performance, demonstrating the generality of our modular framework. Furthermore, in Fig. 6, we observe that the performance of SMN can be further improved by adding new modules. When we use the outputs from module processor for classification, integrating a small number of new modules improve the adaptation capability. We argue this is because new modules focus on distinct parts of images, which are complementary with existing modules.

## 5 CONCLUSION

We propose a novel Scalable Modular Network (SMN) framework, which enables adaptive learning capability and supports integration of new modules after pre-training for better adaptation. These capabilities come from the novel design of agreement router in SMN, which improves the module selection with consideration of both local and global interactions. Experiments demonstrate that SMN exhibits a better generalization capability and supports integration of new modules after pre-training. For future research, it is promising to train a large-scale modular network which supports the integration of new modules for better adaptation. This property enables to adapt general knowledge to specific downstream tasks without forgetting the learned knowledge from pre-trained data.

ACKNOWLEDGMENTS

This work is partially supported by National Key R&D Program of China no. 2021ZD0111901, and National Natural Science Foundation of China (NSFC): 62376259 and 62276246.

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

CONTENTS

## A   THEORETICAL PROOFS

**Theorem 1.** *Let $\{s_1, ..., s_N\}$ be N input vectors and $\{u_{ij} = f_j(s_i)\}_{j=1}^{M}$ a set of output vectors of M modules $\{f_1, ..., f_M\}$. As for the agreement routing alternating between Eq. 1 and Eq. 2, if any two input vectors are not identical, the coefficient $c_{ij} \to 0$ or $1$ as the number of iterations $T \to \infty$.*

*Proof.* With $T + 1$ iterations, the output vector $v_j$ of module $f_j$ can be expanded as

$$v_j^{(T+1)} = \sum_{i=1}^{N} c_{ij}^{(T)} * u_{ij} = \sum_{i=1}^{N} \frac{\exp{(\alpha_{ij}^{(T)})}}{\sum_{j'} \exp{(\alpha_{ij'}^{(T)})}} * u_{ij}. \tag{8}$$

Divide both the numerator and denominator by the term of $\exp{(\alpha_{ij}^{(T)})}$, we have

$$v_j^{(T+1)} = \sum_{i=1}^{N} \frac{1}{1 + \sum_{j' \neq j} \exp{(\alpha_{ij'}^{(T)} - \alpha_{ij}^{(T)})}} * u_{ij}. \tag{9}$$

According to Eq. 1, we have

$$\begin{aligned}
\exp{(\alpha_{ij}^{(T)})} &= \exp{\big(\frac{v_j^{(T-1)} * \boldsymbol{W}_a s_i}{\|v_j^{(T-1)}\| \|\boldsymbol{W}_a s_i\|} + \alpha_{ij}^{(T-1)}\big)} \\
&= \exp{\big(\sum_{t=0}^{T} \frac{v_j^{(t)} * \boldsymbol{W}_a s_i}{\|v_j^{(t)}\| \|\boldsymbol{W}_a s_i\|}\big)}.
\end{aligned} \tag{10}$$

When the iteration number $T \to \infty$, the value of $v_j^{(T)}$ will converge to a fixed vector, denoted as $v_j$, thus the above equation can be rewritten as

$$\exp{(\alpha_{ij}^{(T)})} \approx \exp{\big(T \frac{v_j * \boldsymbol{W}_a s_i}{\|v_j\| \|\boldsymbol{W}_a s_i\|}\big)}, \tag{11}$$

and this result becomes 0 or $\infty$, if the sign of $v_j * \boldsymbol{W}_a s_i$ is negative or positive. Next we prove the theorem by considering two cases: (1) $\alpha_{ij}^{(T)}$ is the largest value among the set $\{a_{ij'}^{(T)}\}_{j'=1}^{M}$. Assume that any two elements in the set $\{s_1, ..., s_N\}$ are not identical, then $\alpha_{ij'}^{(T)} - \alpha_{ij}^{(T)}$ is a negative number for any $j' \neq j$. Thus, the term $\sum_{j' \neq j} \exp{(\alpha_{ij'}^{(T)} - \alpha_{ij}^{(T)})}$ in Eq. 9 tends to 0 and $c_{ij} \to 1$. (2) $\alpha_{ij}^{(T)}$ is not the largest value among the set $\{a_{ij'}^{(T)}\}_{j'=1}^{M}$. In this case, at least one element in set $\{a_{ij}^{(T)}\}_{j=1}^{M}$ is larger than $\alpha_{ij}^{(T)}$. Thus, the term $\sum_{j' \neq j} \exp{(\alpha_{ij'}^{(T)} - \alpha_{ij}^{(T)})} \to \infty$ and $c_{ij} \to 0$. $\qquad\square$

## B    MODULATED MODULES

SMN does not specify the architecture of each module, as the agreement router utilize the agreement between input vectors and the output of each module. Thus, SMN can be implemented with arbitrary module designs, such as MLP, CNN, RNN or self-attention blocks. Inspired by previous works, we utilize a modulated module design in our SMN, which is complementary with existing module architecture and useful to adapt to new tasks with few labeled samples. The modulated modules allows a module to condition its computation on a small set of parameters. As the basis of other modulate module, we define a Modulated Fully-Connected (ModFC) layer as

$$\boldsymbol{y} = \text{ModFC}(\boldsymbol{x}; \boldsymbol{c}) = \boldsymbol{W}(\boldsymbol{x} \odot \text{LayerNorm}(\boldsymbol{W}_c \boldsymbol{c})) + \boldsymbol{b}, \tag{12}$$

where $\odot$ denotes element-wise product, $\boldsymbol{W} \in R^{d_{out}} \times R^{d_{in}}$ is a weight matrix, $\boldsymbol{b}$ is a bias vector, $\boldsymbol{c} \in R^{d_{code}}$ is the conditioning code vector, and $\boldsymbol{W}_c \in R^{d_{in}} \times R^{d_{code}}$ is the conditioning weight matrix that maps code vector into input space. It replaces the traditional fully-connected layer and enables the input vector $x$ to be multiplicatively conditioned by a learnable vector $c$. As FC layers can be stacked to form a MLP, multiple ModFC layers and activation functions can also be stacked to form a Modulated MLP, which can be defined as:

$$\begin{aligned}
\boldsymbol{y} &= \text{ModMLP}(\boldsymbol{x}; \boldsymbol{c}) \\
&= (\text{ModFC}_L(*; \boldsymbol{c}) \circ \text{Activation} \circ ... \circ \text{ModFC}_1(*; \boldsymbol{c}))(\boldsymbol{x})
\end{aligned} \tag{13}$$

In each modulated module, distinct computations can be executed conditioned on unique learnable vectors $c$. Consequently, when new modules are introduced, only the new introduced code vectors need to be fine-tuned to adapt to new tasks. This property allows for sample-efficient fine-tuning on newly introduced modules.

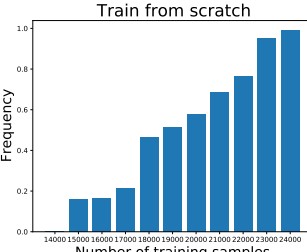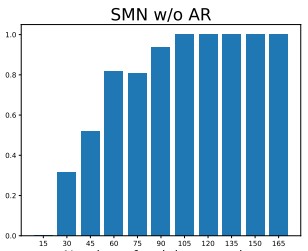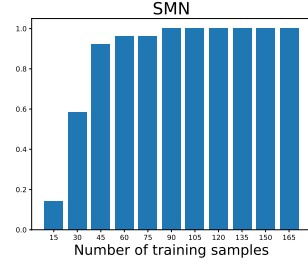

Figure 7: The number of training samples acquired for models to achieve test accuracy of 0.90 on Parity Code task. Left: we train a SMN without agreement router (SMN w/o AR) only on the training data of Parity Code task, which means no knowledge can be pre-acquired. Middle: We pre-train SMN w/o AR on the Min-Max Digit task with 60,000 training data, and then fine-tune it on Parity Code task. Right: We pre-train a SMN and then fine-tune it on the Parity Code task. We run 600 times for each setting with different random seeds and then report the frequency that achieve 0.90 test accuracy on Parity Code task.

## C    EXPERIMENTAL DETAILS

### C.1    IMPLEMENTATION DETAILS FOR MIN-MAX GAME

**Out-of-distribution Generalization.**    We use two convolution layers with ReLU activation function as the tokenizer for SMN. Besides, an average aggregation operation is used in the classifier component to aggregate output vectors from modules and acquire a single vector for classification. For a fair comparison, we use the same convolutional network as backbone to extract features for other models, except the Transformers. We use SGD optimizer to optimize model parameters with learning rate 0.01. In all experiments, we train different models with enough epochs from $\{10, 15, 20\}$ to achieve the training accuracy of 0.99. We set $\lambda = 0.001$ in both the Min-Max Digit task and Parity Code task, and run all experiments 10 times with different random seeds then report the average results with standard deviation.

**Few-shot Adaptation.**    We use the same model architecture as previous experiments. We train these models with 60,000 training images on Min-Max Digit task and fine-tune them with $k$ labeled samples on the Parity Code task. For fine-tuning, we fix the parameters of feature extractor and only fine-tune the classifier (or classification head). For our SMN, when new modules are added, we fix the parameters of tokenizer and existing learned modules, and fine-tune the new added modules with a classifier for adaptation. We use Adam optimizer with learning rate 0.001, and fine-tune each model with 100 epochs.

### C.2    MORE RESULTS FOR MIN-MAX GAME

**Sample Complexity.**    We also explore the task adaptation capability of SMN from the aspect of sample complexity. Fig. 7 shows that for SMN, learning from Min-Max Digit task helps reduce the required number of labeled samples in the Parity Code task (30-45 labeled samples in the right subfigure), greater than the number of samples for the same model but without agreement router (75-105 labeled samples in the middle subfigure), and both outperforms than de novo learning of Parity Code task from scratch (22000-24000 labeled samples in the left subfigure). This comparison shows that the agreement router helps to reduce the sample complexity in the transfer learning scenario.

### C.3    EFFECT OF FINE-TUNING AGREEMENT ROUTER ON MIN-MAX GAME

In the min-max game, when a pre-trained SMN needs to adapt to the parity code task by adding new modules, we fix all existing pre-trained parameters, including the weight $W_a$ in the agreement router, and only fine-tune the newly add modules. We freeze $W_a$, when new modules are introduced, with the consideration of scalability. Because fine-tuning $W_a$ could lead to additional computation load

| Variants | Test Accuracy | | |
|---|---|---|---|
| | add=0 | add=2 | add=4 |
| Freeze $W_a$ | 87.9 (4.1) | 88.7 (2.0) | 88.7 (1.7) |
| Fine-tune $W_a$ | 91.5 (3.8) | 92.6 (2.2) | 89.0 (3.7) |

Table 5: Test accuracy of SMN variants on parity code task with different number of added modules.

| Method | | FLOPs | Training Time | Inference Time | Accuracy |
|---|---|---|---|---|---|
| Top-K | | 7.6M | 140s | 2.3s | 42.94 (2.1) |
| Truncated | | 10.8M | 120s | 2.3s | 43.77 (3.4) |
| | $T = 0$ | 10.8M | 80s | 2.3s | 44.55 (2.2) |
| SMN | $T = 2$ | 12.2M | 130s | 2.6s | 47.60 (4.1) |
| | $T = 4$ | 13.6M | 160s | 2.7s | 54.46 (5.0) |

Table 6: Computational cost analysis of SMN and other comparison methods in the min-max game experiment.

and loss of learned routing information on old tasks. However, when we only focus on the current task, fine-tuning $W_a$ is a good choice to enhance the performance. Here we give more experimental results to show the impact of fine-tuning $W_a$. In the Tab. 5, we adapt the pre-trained SMN to the parity code task by finetuning both $W_a$ and newly added modules. We observe that fine-tuning $W_a$ leads to further performance improvement for this task. We argue this is because fine-tuned agreement router adapts to the parity code task, facilitating the selection of specialist modules.

## C.4 COMPUTATIONAL COST ANALYSIS

For analyzing computational cost, we report the FLOPs, training and inference time (seconds), and accuracy for our SMN and other modular variants in the min-max game experiment. The training time is calculated on 60,000 images with 10 epochs, and the inference time is calculated on 20,000 test images. We acquire these statistical information based on a single 3090 GPU.

Form Tab. 6, We can find that when the iteration number $T = 2$, SMN exhibits comparable training and inference times to Top-K and Truncated methods, while outperforms them in term of accuracy. Notably, increasing iteration number $T$ can improve the performance of SMN further, while bringing more computational burden.

## C.5 DETAILS FOR FEW-SHOT IMAGE CLASSIFICATION

**Dataset Details** We consider three popular benchmark in FSIC task: *mini*ImageNet (Vinyals et al., 2016), CIFAR-FS (Bertinetto et al., 2018) and CUB-200-2011 (CUB) (Wah et al., 2011). *mini*ImageNet is a subset of ImageNet (Russakovsky et al., 2015) consisting of 60,000 images uniformly distributed over 100 object classes. The train/validation/test splits consist of 64/16/20 object classes, respectively. CIFAR-FS is derived from CIFAR100 (Krizhevsky et al., 2009), and it contains 100 classes with class split as 64, 16, and 20. CUB is a fine-grained bird classification dataset, consisting of 100/50/50 bird classes for train/validation/test splits.

**Experiment Setup.** As most previous works (Finn et al., 2017; Snell et al., 2017; Chen et al., 2019), we use the $N$-way $K$-shot setting to construct training and test episodes from dataset. It means that we first randomly choose $N$ classes from dataset, then choose $K$ support samples and $Q$ query samples for each class to construct a few-shot episode. In each episode, a few-shot model is designed to classify these $N \times Q$ query images into $N$ classes based on $N \times K$ support samples. Note that the classes of training episodes $C_{train}$ will not overlap with the classes of test episodes $C_{test}$, which means $C_{train} \bigcap C_{test} = \emptyset$. More specifically, for the 5-way 1-shot (and 5-shot) evaluation, we test 15 query samples for each class in an episode and report the average accuracy with 95% confidence intervals of randomly sampled 600 test episodes.

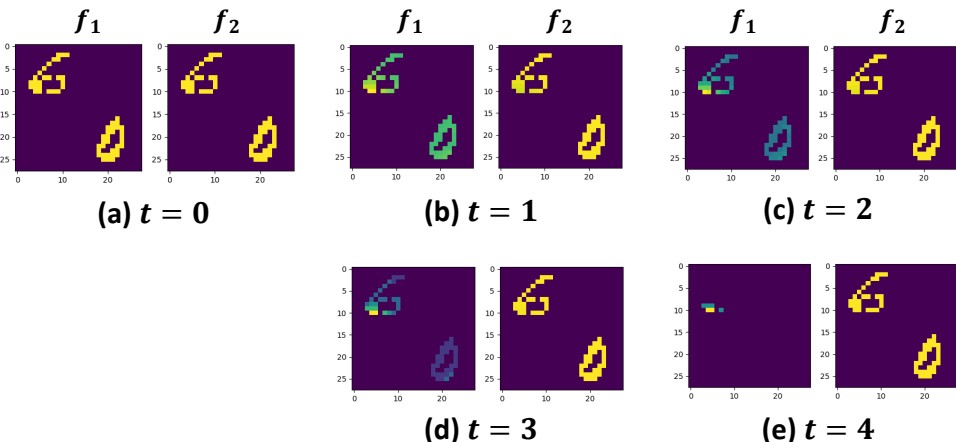

Figure 8: Activation map of two modules within the SMN ($T = 4$). We visualize the activation map at each iteration on an example from composition (6,0).

**Implementation Details.** Note that our Scalable Modular Network do not specify the choice of tokenizer, any set-valued network that output a set of vectors is feasible. We adopt the following two popular backbones as our tokenizer: (a) Conv4-64 (Vinyals et al., 2016), a convolutional neural network which consists of 4 convolution layers with 64/64/64/64 filters for a total of 0.113M parameters; (b) ViT-S (Lin et al., 2023), a small size vision transformer which consists of 12 transformer encoders for a total of 21M parameters. For the former choice, we train the conv4-64 with module processor and classifier from scratch. For the latter, we first pre-train the vision transformer and projection head with a self-supervised loss following (Lin et al., 2023). After that, the pre-trained model is used as our tokenizer, and we train it with other parts by the combined loss in Eq. 6.

## D    VISUALIZATION OF ITERATIVE DYNAMICS

To better understand what occurs in each iteration for the agreement router, We visualize the activation map of two modules $f_1$ and $f_2$ per iteration from a trained SMN model. As shown in Fig 8, both modules $f_1$ and $f_2$ are activated in response to the input image at the initial stage ($t = 0$). As the iteration proceeds, module $f_1$ gradually deactivates, while module $f_2$ remains unchange. In the iterative dynamics, the input-output agreement can be treated as if it was a log-likelihood and is added to the previous logit $a_{ij}^{(t-1)}$, as expressed in Line 3 of Algorithm 1. Guided by the accumulated loglikelihood $\{a_{ij}^{(t)}\}_{j=1}^{M}$, the router models a conditional distribution to *classify* which specalist modules can acquire the input vector $s_i$ (as expressed in Line 4 of Algorithm 1), although in fact, there does not exist ground-truth labels for the classification task.

## E    EVALUATION ON DIVERSE TASKS AND DATASETS

We conduct additional experiments to demonstrate the generality and effectiveness of our SMN in diverse tasks and datasets, including reinforcement learning, continual learning and cross-domain few-shot learning tasks.

### E.1    REINFORCEMENT LEARNING.

We consider a reinforcement learning task of path finding from openai gym (Chevalier-Boisvert et al., 2018) for evaluating the out-of-distribution generalization ability of models. More specifically, in the path finding task, an agent must open a door using a key and then get to the goal with only partial observation of the environment. See an example of the path finding task within a 8x8 room at Fig 9. We train an RL agent using Proximal Policy Optimization (PPO) (Schulman et al., 2017) building upon the representations from a modular network. The training environment is set within a

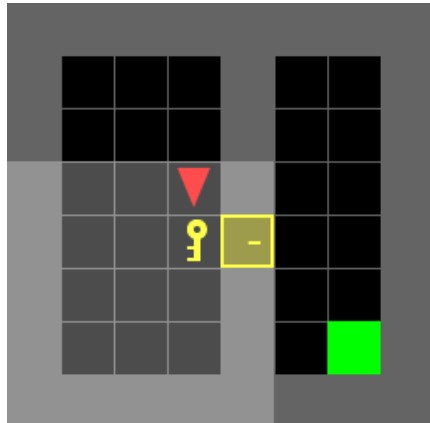

Figure 9: An example of the path finding task within a 8x8 room. The red triangle represents an agent, and the green square is the goal. Yellow square is a locked door, which should be opened with a key.

5x5 room, while evaluations are conducted in different environments with 6x6 and 8x8 rooms. From Tab. 7, we can find that SMN performs better than other methods, and the agreement router plays an important role in the generalization ability of SMN.

| Method | | 5x5 Room | 6x6 Room | 8x8 Room |
|---|---|---|---|---|
| Top-K | | 0.88 (0.05) | 0.88 (0.05) | 0.80 (0.16) |
| Truncated | | 0.88 (0.05) | 0.88 (0.05) | 0.74 (0.18) |
| | $T = 0$ | 0.81 (0.09) | 0.81 (0.09) | 0.71 (0.22) |
| SMN | $T = 2$ | 0.88 (0.05) | 0.88 (0.05) | 0.82 (0.20) |
| | $T = 4$ | **0.91 (0.03)** | **0.91 (0.03)** | **0.85 (0.18)** |

Table 7: Results of SMN and other methods in the path finding task. We scored different models by computing the average total reward of the last 100 episodes.

| | min-max | parity |
|---|---|---|
| add=0 | 96.12 (0.13) | 87.48 (3.98) |
| add=1 | 95.42 (0.35) | 88.61 (3.31) |
| add=2 | 94.50 (1.73) | 89.32 (2.34) |

| | min-max | parity |
|---|---|---|
| add=0 | 78.17 (5.27) | 91.5 (3.8) |
| add=1 | 75.19 (6.47) | 92.2 (3.3) |
| add=2 | 73.57 (5.51) | 92.6 (2.2) |

Table 8: Performance of SMN on min-max digital task (**min-max**) and parity code task (**parity**) with different number of added modules.

Table 9: Performance of SMN on two tasks. We fine-tune the pre-trained $W_a$ and newly added modules together.

### E.2 CONTINUAL LEARNING.

We also construct a continual learning scenario to verify the adaptation ability of SMN without forgetting previous knowledge. In this scenario, a min-max digital task and parity code task comes sequentially. The first task is comprised with sufficient training data, while the second task only contains limited training data. Solving this problem with SMN is straightforward by applying different specialist modules based on the task ID, as parameter-isolation methods in task incremental learning (Rusu et al., 2016). However, we explore a more difficult setting to demonstrate the scalabity of agreement router, where the task ID is unavailable in the module selection process. More specifically, We train SMN with 2 modules on the first task, and then adapt it to the second task by adding 1-2 new modules. During test, the agreement router cannot acquire the task information to select specialist modules. Tab. 8 shows that introducing new modules for the parity code can improve the performance, meanwhile only causing a little decrease in the min-max task. We attribute

this to the scalablity of agreement router, as we observe that agreement router adaptively deactivate the newly added modules for the old task. Moreover, we explore the impact of fine-tuning $W_a$ on performance. Tab. 9 show that fine-tuning $W_a$ on the parity code task will hurt the performance in the min-max digital task, as the learned routing information on old task loses.

| Method | Backbone | *mini*ImageNet→CUB | | *mini*ImageNet→Sketch | |
|---|---|---|---|---|---|
| | | 5-way 1-shot | 5-way 5-shot | 5-way 1-shot | 5-way 5-shot |
| ProtoNet | | 33.91 (0.67) | 53.74 (0.72) | 41.05 (0.76) | 63.26 (0.72) |
| RelationNet | | 38.19 (0.69) | 52.57 (0.66) | 46.33 (0.81) | 59.46 (0.74) |
| MAML | Conv4 | 36.97 (0.69) | 51.60 (0.70) | 42.95 (0.83) | 58.51 (0.78) |
| Baseline++ | | 37.11 (0.66) | 52.42 (0.67) | 36.46 (0.75) | 51.79 (0.77) |
| SMN | | **42.14 (0.72)** | **61.20 (0.73)** | **48.33 (0.86)** | **69.48 (0.75)** |

Table 10: 5-way 1-shot and 5-shot performance of different FSL methods on cross-dataset scenario (*mini*ImageNet→CUB and *mini*ImageNet→Sketch). Conv4-64 is used as the backbone model. We report the average accuracy on 600 novel tasks with 95% confidence interval.

### E.3 CROSS-DOMAIN FEW-SHOT LEARNING

To demonstrate the effectiveness of our SMN in more challenging tasks, we consider the cross-domain few-shot learning problem, where the source and target domains are dissimilar.

**Datasets and Setup.** Based on the setup of Section 4.2, we consider two cross-domain few-shot learning scenario: (1) from *mini*Imagenet to CUB (*mini*Imagenet→CUB), as miniImageNet is a general object recognition dataset and CUB is a fine-grained classification dataset; (2) from *mini*Imagenet to ImageNet-Sketch (*mini*Imagenet→Sketch). ImageNet-Sketch (Wang et al., 2019) contains 50,000 sketch images belonging to the same 1,000 ImageNet classes, but exists large domain gap with ImageNet images. For the former scenario, we use 100 classes of miniImageNet as training classes, and the 50 validation and 50 test classes from CUB. For the latter, we also 100 classes of miniImageNet as training classes, and 900 classes from ImageNet-Sketch to keep the training and novel classes non-overlapped.

**Results.** Tab. 10 shows that our SMN owns a better adaptation ability than other FSL methods in both two cross-domain scenarios, which demonstrates the adaptation and out-of-distribution generalization ability of SMN.

