# OpenReview forum: "Scalable Modular Network: A Framework for Adaptive Learning via Agreement Routing"
_ICLR.cc/2024/Conference — ICLR 2024 poster_

### Official Review · Reviewer_Zqvo · 2023-10-13

**Soundness:** 3 good
**Presentation:** 3 good
**Contribution:** 3 good
**Rating:** 6
**Confidence:** 4

**Summary:**

This paper presents Scalable Modular Network for better adaptive learning by incorporating new modules after pre-training. An agreement router is proposed to select specialist modules using an iterative message passing process. The approach is evaluated with a min-max game task and few-shot image classification task.

**Strengths:**

1. Modular networks have certain advantages in some machine learning settings such as meta-learning and continual learning.
2. The proposed agreement router is novel and effective.

**Weaknesses:**

1. Some evaluation under continual learning setting may be desirable. Evaluation in the paper is not sufficiently strong with one toy task and one few shot learning setting.

**Questions:**

1. Fix typos, e.g, "impose constrains on the number of activated..."

---

> ### Author Response · Authors · 2023-11-20
> **Rebuttal by Authors**
>
> Thank you for taking the time to review our work.
> We are glad that you deem the agreement router to be novel and effective, and the modular networks advantageous.
> In the following, we address your questions and concerns.
>
> **W1: Some evaluation under continual learning setting may be desirable. Evaluation in the paper is not sufficiently strong with one toy task and one few shot learning setting.**
>
> Thank you for this suggestion.
> We try to verify the generality of SMN in the following three diverse tasks:
> + **Continual Learning.** We consider a continual learning scenario to verify the adaptation ability of SMN without forgetting previous knowledge. In this scenario, the min-max digital task and parity code task come sequentially.
> The first task comprises with sufficient training data, while the second task only contains limited training data.
> Detailed experimental setting and results can be found in the response to Question 8 for the Reviewer nZgT.
> The experimental results demonstrate that SMN can adapt to new task better with new modules incorporated, without forgetting the old tasks.
> + **Cross-Domain Few-Shot Learning**. To demonstrate the adaptation ability of our SMN in more chanllenging tasks, we delve into the cross-domain few-shot learning (CD-FSL) scenario.
> In this context, the source and target domains are dissimilar, thereby intensifying the difficulty of adapting to novel tasks.
> We conduct experiments on two setting: miniImageNet --> CUB and miniImageNet --> ImageNet-Sketch.
> See the experimental results in our response to Weakness 1 for Reviewer y9sx. We find that our SMN owns a better adaptation ability than other FSL methods, even in the more chanllenging scenario.
> + **Reinforcement Learning.** We consider a reinforcement learning task of path finding to evaluate the out-of-distribution generalization ability of SMN.
> More specifically, in the path finding task, an agent must open a door by a key and then get to the goal with only partial observation of the environment.
> We train our SMN and other compared methods in the 5x5 room but evaluate them on the 6x6 and 8x8 rooms.
> The detailed experimental setting and results can be founded in the response to Weakness 1 of Reviewer 9n4d.
> We find that SMN performs better than other methods, and the agreement router plays an important role in the generalization ability of SMN.
>
> **Q1: Fix typos, e.g, "impose constrains on the number of activated..."**
>
> Thanks for pointing out this. It should be "impose constraints...".
> We have fixed it in the updated version.

---

> > ### Comment · Reviewer_Zqvo · 2023-11-22
> > **Thanks for the authors' response.**
> >
> > Based on the authors' response, my rating remains the same.

---

### Official Review · Reviewer_y9sx · 2023-10-30

**Soundness:** 3 good
**Presentation:** 2 fair
**Contribution:** 3 good
**Rating:** 6
**Confidence:** 3

**Summary:**

The paper introduces the Scalable Modular Network (SMN), a modular framework with adaptive learning capabilities that can incorporate new modules after initial training for better adaptation.

**Key Features:**
- **Agreement Router:** A unique component in SMN that iteratively selects and assembles specialist modules based on both local and global interactions.

- **Dynamic Module Selection:** Allows SMN to adjust module combinations adaptively based on input data.

- **Scalability:** Enables the addition of new modules post-training, enhancing adaptability.

**Benefits:**
SMN efficiently selects modules for new samples, generalizes for out-of-distribution data, and can differentiate between similar sub-concepts using global information.

**Experiments:**
Tests on a toy min-max game and few-shot image classification demonstrated SMN's adaptive capabilities, especially when adding new modules post-training.

**Significance:**
SMN offers a solution to the challenge of composing specialist modules in neural networks, moving closer to achieving human-like learning efficiency in machines.

**Strengths:**

1. **Agreement Router:** Introduces a dynamic module selection mechanism, mirroring human cognitive abilities for efficient and adaptive learning.

2. **Scalability:** Allows for the integration of new modules post-training, ensuring the network's adaptability and evolution.

**Weaknesses:**

1. **Limited Experimentation:** The experiments are overly simplistic, conducted only on small datasets.

2. **Lack of Broad Testing:** Results focus solely on classification tasks, with no results provided for large language models like LMM.

3. **Additional Fine-tuning:** The addition of new modules still requires extra fine-tuning, implying integration isn't as seamless as desired.

4. **No Comparison with MOE:** The paper doesn't offer a comparison with established methods like MOE, limiting the understanding of its relative performance.

5. **Efficiency Overlooked:** There's no comparison or discussion regarding computational efficiency or time delays, making it hard to assess the practicality of deployment.

**Questions:**

See the weakness.
Additional Question:

1. **Pre-training Paradigm:** The abstract mentions the applicability of SMN in pre-training paradigms, but there seems to be limited experimental evidence or discussion on this. How does the SMN truly perform in a pre-training context?

2. **Comparison with Current Techniques:** Modern pre-training often employs methods like contrastive learning or MLM (Masked Language Model) loss. Is there a significant gap between SMN and these prevalent techniques? How does SMN align or differentiate from these established methods?

---

> ### Author Response · Authors · 2023-11-20
> **Rebuttal by Authors (1/2)**
>
> Thank you for taking the time to review our work.
> We are glad that you were interested in the scalability of our method, and we appreciate your suggestions to enhance the impact of our paper.
> In the following, we address your questions and concerns in the order of being proposed.
>
> **W1: The experiments are overly simplistic, conducted only on small datasets.**
>
> To demonstrate the effectiveness of our SMN in more chanllenging tasks, we consider the cross-domain few-shot learning problem, where the source and target domains are dissimilar.
> Following [ref1], we consider a cross-domain scenario from  miniImageNet to CUB, as miniImageNet is a general object recognition dataset and CUB is a fine-grained classification dataset.
> Besides, we consider another cross-domain scenario from miniImageNet to ImageNet-Sketch [ref2], where we select a subset of classes from ImageNet-Sketch to keep the training and novel classes non-overlapped.
> The results are as follow:
>
> + **miniImageNet -> CUB**
>   | Method       | 5-way 1-shot | 5-way 5-shot |
>   |--------------|-------------|----------|
>   | MatchingNet  |  37.48 (0.68)   | 49.98 (0.66)   |
>   | ProtoNet     | 33.91 (0.67)    | 53.74 (0.72)   |
>   | RelationNet  | 38.19 (0.69)    | 52.57 (0.66)   |
>   | MAML         | 36.97 (0.69)    | 51.60 (0.70)   |
>   | Baseline++   | 37.11 (0.66)    | 52.42 (0.67)   |
>   | SMN          | **42.14 (0.72)**    | **61.20 (0.73)**   |
>
> + **miniImageNet -> ImageNet-Sketch**
>   | Method       | 5-way 1-shot | 5-way 5-shot |
>   |--------------|-------------|----------|
>   | MatchingNet  | 44.38 (0.78)    | 60.76 (0.74)   |
>   | ProtoNet     | 41.05 (0.76)    | 63.26 (0.72)   |
>   | RelationNet  | 46.33 (0.81)    | 59.46 (0.74)   |
>   | MAML         | 42.95 (0.83)    | 58.51 (0.78)   |
>   | Baseline++   | 36.46 (0.75)    | 51.79 (0.77)   |
>   | SMN          | **48.33 (0.86)**    | **69.48 (0.75)**   |
>
> We find that our SMN owns a better adaptation ability than other FSL methods, even in the chanllenging cross-domain scenario.
>
> [ref1] A Closer Look at Few-shot Classification. ICLR 2019.
>
> [ref2] Learning robust global representations by penalizing local predictive power. NeurIPS 2019.
>
> **W2: Results focus solely on classification tasks, with no results provided for large language models like LMM.**
>
> We apply SMN in diverse applications, including reinforencement learning and continual learning tasks.
> The experimental details and results for these two tasks are provided in the response to Weakness 1 of Reviewer 9n4d.
> The results across diverse tasks demonstrate the generality of our SMN and the effectiveness of the agreement router.
> As for building a large-scale modular network, such as those seen in large language models, is outside the scope of this work, we leave it into future exploration.
>
> **W3: The addition of new modules still requires extra fine-tuning, implying integration isn't as seamless as desired.**
>
> Although incorporating new modules requires extra fine-tuning to adapt to new tasks, a limited number of labled samples proves sufficient for this fine-tuning process.
> In the toy min-max game, we fine-tune the newly added modules using only 1-6 labeled samples per composition.
> Similarly, in few-shot image classification, we fine-tune the newly incorporated modules with just 1 or 5 labeled samples per class.
> This sample-efficient adaptation ability stems from the design of SMN framework, where the most parameters of new modules (e.g., $W$ and $W_c$ in Equation 12) is shared with pre-trained modules, as introduced in the Section 3.5 and Appendix B.
>
> **W4: The paper doesn't offer a comparison with established methods like MOE, limiting the understanding of its relative performance.**
>
> We have provided a fair comparison between our SMN and MoE in both Table 1 and Table 2 of main paper.
> The compared Top-K method follows the same training strategy as MoE, which contains a noisy top-$k$ gating mechanism and an importance loss during training.
> For a fair comparison, we adopt the same module structure as employed in our SMN for the Top-K method, and maintain the same number of modules.
> The results presented in Table 1 and Table 2 demonstrate that our SMN can outperforms Top-K method with the same number of specialist modules.
>
> **W5: There's no comparison or discussion regarding computational efficiency or time delays, making it hard to assess the practicality of deployment.**
>
> For computational cost, we report the FLOPs, training and inference time (seconds), and accuracy for our SMN and other modular variants in the min-max game experiment.
> The details and observations can be founded in our response to Weakness 3 & Question 7 for Reviewer nZgT.
> We can find that when the iteration number $T=2$, SMN exhibits comparable training and inference times to Top-K and Truncated methods, while outperforms them in term of accuracy.
> Notably, increasing iteration number $T$ can improve the performance of SMN further, while bringing more computational burden.

---

> ### Author Response · Authors · 2023-11-20
> **Rebuttal by Authors (2/2)**
>
> **Q1: There seems to be limited experimental evidence or discussion on pre-training paradigms. How does the SMN truly perform in a pre-training context?**
>
> We have discussed the pre-training paradigms when evaluating the fast adaptation ability of SMN: initially pre-training a model on some base tasks with sufficient data and subsequently fine-tuning it on novel tasks with limited data.
> More specifically, in the toy min-max game, SMN performs pre-training on the min-max digital task, followed by fine-tuning in the parity code task. Similarly, in the few-shot image classification task, the process begins with pre-training SMN on base training tasks, then fine-tuning on novel tasks.
> Experimental results demonstrate the adaptation ability of SMN in the pre-training paradigm.
>
> **Q2: Is there a significant gap between SMN and contrastive learning/masked language model loss? How does SMN align or differentiate from these established methods?**
>
> The SMN framework is complementary with existing self-supervised techniques, such as mask image modeling or contrastive learning.
> Given that the specialist modules and agreement router build upon the tokenizer, self-supervised techniques can be employed to initialize the tokenizer effectively, thereby enhancing module specialization.
> In few-shot image classification experiments, we adopt a two-stage pre-training process following [ref].
> Initially, we pre-train a ViT-S backbone with a mask image modeling loss as our tokenizer. Subsequently, we pre-train the entire SMN model on training tasks using both supervised loss and an importance loss.
> We empirically find that using the self-supervised technique can significantly improves the performance of SMN model.
>
> [ref] Supervised masked knowledge distillation for few-shot transformers. CVPR 2023.

---

### Official Review · Reviewer_9n4d · 2023-10-31

**Soundness:** 3 good
**Presentation:** 3 good
**Contribution:** 3 good
**Rating:** 6
**Confidence:** 2

**Summary:**

This paper presents an innovative modular network framework, the Scalable Modular Network (SMN), which represents a significant advancement in the realm of adaptive learning. SMN not only enhances the capacity for adaptive learning but also offers a seamless mechanism for integrating new modules post pre-training, thus markedly improving overall adaptability. This adaptive process is underpinned by the ingenious "agreement router" integrated within SMN, which streamlines the module selection process by carefully weighing both local and global interactions. Throughout the experimental validation on two distinct problems, a toy min-max game, and a few-shot image classification task, SMN consistently showcases remarkable generalization abilities to new data distributions and demonstrates sample-efficient adaptation to novel tasks. Notably, SMN's adaptation capabilities are further enhanced when new modules are introduced after pre-training. These results underscore the potential of the SMN to continually evolve and improve its adaptive capabilities.

**Strengths:**

1. This paper effectively presents the motivation behind the study and articulates the issues addressed by the proposed method in a well-structured manner.
2. The authors provide a comprehensive review of existing methods related to modular neural networks. They also meticulously analyze the drawbacks and issues that need resolution in these existing approaches.
3. The Scalable Modular Network (SMN) is presented as a concise and efficient method with remarkable clarity in both its rationale and technical details. The paper offers ample explanations, discussions, and evidence to underpin the theoretical foundation of SMN.
4. The inclusion of implementation details and experimental settings further substantiates the fairness of comparisons across two distinct problems.
5. Moreover, the study hints at the significant potential value in large-scale modular networks capable of seamlessly integrating additional modules.

**Weaknesses:**

1. The experiments conducted on a toy min-max game and few-shot image classification may not be sufficient to definitively establish the overall superiority of SMN, considering that SMN is positioned as a general framework. Additional experiments on a wider range of datasets and tasks would better illustrate the effectiveness of the method
2. The paper falls short in providing adequate explanations of limitations and training costs.

**Questions:**

Please kindly find the comments in the weakness section.

---

> ### Author Response · Authors · 2023-11-20
> **Rebuttal by Authors**
>
> Thank you for taking the time to review our work.
> We are glad that you recognized the potential value of our work in large-scale modular network, and deemed our modular framework to be innovative and ingenious.
> In the following, we address your questions and concerns in the order of being proposed.
>
> **W1: Additional experiments on a wider range of datasets and tasks would better illustrate the effectiveness of the method.**
>
> Thanks for this valuable suggestion.
> We try to verify the generality of SMN in the following diverse tasks:
> + **Reinforcement Learning.** We consider a reinforcement learning task of path finding from openai gym [ref1] for evaluating the out-of-distribution generalization ability of models.
> More specifically, in the path finding task, an agent must open a door using a key and then get to the goal with only partial observation of the environment.
> See an example of the path finding task in Figure 9 at Appendix E.1.
> We train an RL agent using Proximal Policy Optimization (PPO) [ref2] building upon the representations from a modular network.
> The training environment is set within a 5x5 room, while evaluations are conducted in different environments with 6x6 and 8x8 rooms.
>   The results are as follows:
>
>   |      | 5x5 room | 6x6 room | 8x8 room |
>   |--------------|-------------|----------|-------------|
>   | Top-K       | 0.88 (0.05) | 0.88 (0.05)    | 0.80 (0.16)   |
>   | Truncated   | 0.88 (0.05)    | 0.88 (0.05)   | 0.74 (0.18)   |
>   | SMN ($T=0$) | 0.81 (0.09)    | 0.81 (0.09)   | 0.71 (0.22)   |
>   | SMN ($T=2$) | 0.88 (0.05)    | 0.88 (0.05)   | 0.82 (0.20)   |
>   | SMN ($T=4$) | **0.91 (0.03)**    | **0.91 (0.03)**   | **0.85 (0.18)**   |
>
>   In the table above, we scored different models by computing the average total reward of the last 100 episodes, as previous RL works [ref2].
>   We can find that SMN performs better than other methods, and the agreement router plays an important role in the generalization ability of SMN.
>
>   [ref1] Minimalistic gridworld environment for openai gym. 2018
>
>   [ref2] Proximal policy optimization algorithms. Arixv 2017.
> + **Continual Learning.** We also construct a continual learning scenario to verify the adaptation ability of SMN without forgetting previous knowledge. In this scenario, a min-max digital task and parity code task comes sequentially.
> The first task is comprised with sufficient training data, while the second task only contains limited training data.
> Detailed experimental setting and results can be found in the response to Question 8 for the Reviewer nZgT.
> From the experimental results, we demonstrate that SMN can adapt to new task better with incorporated with new modules, without forgetting the old tasks.
> + **Cross-Domain Few-Shot Learning**. To demonstrate the adaptation ability of our SMN in more chanllenging tasks, we delve into the cross-domain few-shot learning (CD-FSL) scenario.
> In this context, the source and target domains are dissimilar, thereby intensifying the difficulty of adapting to novel tasks.
> We conduct experiments on two setting: miniImageNet --> CUB and miniImageNet --> ImageNet-Sketch.
> See the experimental results in our response to Weakness 1 for Reviewer y9sx. We find that our SMN owns a better adaptation ability than other FSL methods, even in the more chanllenging scenario.
>
> **W2: The paper falls short in providing adequate explanations of limitations and training costs.**
>
> Thank you for pointing out this.
> Here we will explain the training costs and limitations of our work:
>
> + **Train costs**: The training time of SMN in the toy min-max game is 160 seconds on a single 3090 GPU, and in the few-shot image classification task is 3 hours on four 3090 GPUs.
> More details about the computational cost analysis can be found in our response to Weakness 3 & Question 7 for Reviewer nZgT.
> Notably, increasing iteration number $T$ can improve the performance of SMN further, while bringing more computational burden.
> + **Limitations**: Agreement router needs to activate all modules and then iteratively calculates the input-output agreement, which increases the computational burden. Moreover, in this work, we
> demonstrate the effectiveness of our SMN on small size of datasets.
> However, we believe our work lays a foundation for adaptive learning, offering a start point for further research to train a large-scale modular network.

---

### Official Review · Reviewer_nZgT · 2023-11-02

**Soundness:** 3 good
**Presentation:** 3 good
**Contribution:** 3 good
**Rating:** 6
**Confidence:** 4

**Summary:**

The paper presents a novel approach for scalable modular framework. They propose a unique method for learning routing among a set of modules by averaging outputs across input tokens, computing agreement with each of the input tokens, refining the outputs based on this agreement, and finally aggregating these refined outputs for final  classification. The method showcases adaptability to a new task (parity code task) after training on a different task (min-max detection). However, the scalability and adaptation mechanisms, especially when new modules are introduced or when there are multiple layers of modules, are not clearly explained.

**Strengths:**

1. The method displays remarkable results on a toy dataset and adapts well to new tasks (parity code task)
2. Better performance on the ConvNet backbone for few-shot classification task on real-world datasets as compared to previous methods.

**Weaknesses:**

1. The scalability of the proposed method is not clearly demonstrated, especially regarding how weights W_a in agreement router and W_c  in module parameters are adapted with the introduction of new modules.
2. The paper proposes a method that works on a single layer, which is quite restrictive. But doesn't dive deeper into how the method translates when using multiple layers of modules.
3. A detailed computational cost analysis is missing, which is crucial for understanding the trade-offs, especially since the proposed method seemingly demands more compute by activating all modules repeatedly.
4. The performance on the ViT backbone does not mirror the improvements seen on the ConvNet backbone, raising questions on the consistency of the method's performance across different backbones.
5. The method needs additional losses terms to prevent degenarcy similar to Top-K approaches.

**Questions:**

1. How is \(W_a\) adapted when new modules are introduced, and how does this adaptation impact the performance ?
2. Can you explain the relationship between \(y\) from Equation 4 and the outputs from the modules?
3. How does the method work when there are multiple layers of modules?
4. In Equation 12, is W_c frozen or trainable when new modules are added?
5. With only two modules, what is the value of K in the Top-K method in Table 1, and how does this compare to an Ensemble approach where all modules are activated?
6. Could you provide more insight on what occurs in every iteration of the proposed method, and elaborate on the notion of agreement in this context?
7. Could you add more on the trade-off between computational cost and accuracy of the proposed method vs prior methods?
8. Is there a plan to conduct experiments showcasing the scalability of the method, particularly when adapting to a new task by adding new modules without forgetting old tasks? You can choose the same setup of learning min-max detection and parity code task one after the other by adding new modules.

---

> ### Author Response · Authors · 2023-11-20
> **Rebuttal by Authors (1/3)**
>
> Thank you for taking the time to review our work.
> We are glad that you deemed our method to be novel and unique.
> We appreciate the valuable questions and concerns, and try to address them in the following.
>
> **W1&Q1&Q4: How the weight $W_a$ of agreement router (in Equation 1) and $W_c$ in module parameters (in Equation 12) are adapted when new modules are introduced? And, how does this adaptation impact the performance?**
>
> In the SMN framework, we freeze both the pre-trained weight $W_a$ and $W_c$ when introducing new modules.
> We explain the reason of this setting and show how it impact the performance.
> + **Freeze $W_c$ in module parameters.** In our SMN framework, most parameters of new modules (e.g., $W$ and $W_c$ in Equation 12) are shared with  pre-trained modules.
> In this case, fine-tuning the pre-trained weight $W_c$ will destory the learned knowledge within existing modules, leading to a decline in performance, as illustrated in the table below.
>
>   | Variants     | add=0 | add=2 | add=4 |
>   |--------------|-------------|----------|----------|
>   | Freeze $W_c$ | 87.9 (4.1)   | 88.7 (2.0)    | 88.7 (1.7) |
>   | Finetune $W_c$ | 84.9 (3.8) | 85.6 (2.2)    | 85.0 (3.7) |
> + **Freeze $W_a$ of agreement router.** With the consideration of scalability, we freeze $W_a$ when new modules are introduced, because finetuning $W_a$ will lead to additional computation load and loss of learned knowledge (see this result in the response to Question 8).
>
>   | Variants     | add=0 | add=2 | add=4 |
>   |--------------|-------------|----------|----------|
>   | Freeze $W_a$ | 87.9 (4.1)   | 88.7 (2.0)    | 88.7 (1.7) |
>   | Finetune $W_a$ | 91.5 (3.8) | 92.6 (2.2)    | 89.0 (3.7) |
>
> However, when we only focus on the current task, fine-tuning $W_a$ is a good choice to enhance the performance. In the table above (last row), we adapt the pre-trained SMN to the parity code task by finetuning both $W_a$ and newly added modules.
>   We observe that fine-tuning $W_a$ leads to further performance improvement for this task.
>   We argue this is because fine-tuned agreement router adapts to the parity code task, facilitating the selection of specialist modules.
>
> **W2&Q3: How does the SMN work when there are multiple layers of modules?**
>
> To verify the efficacy of SMN in presence of multiple module layers, we stack two layers of modules in the module processor, comprising 8 and 2 specialist modules for each layer, respectively.
> In the multi-layer SMN, each module layer owns a unique agreement router for selecting different specialist modules.
> We evaluate the generalization ability of the multi-layer SMN on the new compositions, following the same setting of Table 1 in the main paper.
> The results are presented below.  We find that stacking more layers of module can further improves the out-of-distribution generalization ability of SMN.
>
> |      | Params | Accuracy |
> |--------------|-------------|----------|
> | $T=4$, Layer=1 | 131.9K       | 54.66 (3.2)    |
> | $T=0$, Layer=2 | 206.2K       | 46.50 (3.5)    |
> | $T=2$, Layer=2 | 206.2K       | 54.99 (2.7)    |
> | $T=4$, Layer=2 | 206.2K       | **56.59 (2.9)**    |
>
>
> **W3&Q7: A detailed computational cost analysis is missing, which is crucial for understanding the trade-offs between cost and accuracy of proposed method vs prior methods.**
>
> Thank you for pointing out this. For analyzing computational cost, we report the FLOPs, training and inference time (seconds), and accuracy for our SMN and other modular variants in the min-max game experiment.
>
> |      | FLOPs | Training Time | Inference Time| Accuracy|
> |--------------|-------------|----------|-------------|----------|
> | Top-K       |  7.6M    | 140   | 2.3   | 45.17 (2.4)    |
> | Truncated   | 10.8M    | 120   | 2.3   | 48.70 (3.4)    |
> | SMN ($T=0$) | 10.8M    | 80    | 2.3   | 46.76 (2.4)    |
> | SMN ($T=2$) | 12.2M    | 130   | 2.6   | 53.95 (1.7)    |
> | SMN ($T=4$) | 13.6M    | 160   | 2.7   | 54.66 (3.2)    |
>
> We can find that when the iteration number $T=2$, SMN exhibits comparable training and inference times to Top-K and Truncated methods, while outperforms them in term of accuracy.
> Notably, increasing iteration number $T$ can improve the performance of SMN further, while bringing more computational burden.

---

> ### Author Response · Authors · 2023-11-20
> **Rebuttal by Authors (2/3)**
>
> **W4: Why the performance on ViT backbone does not mirror the improvements seen on ConvNet backbone?**
>
> In Table 1 and Table 2 of the main paper, the performance improvement gap between ViT and ConvNet backbone can be attributed primarily to two factors:
> + **Stronger backbone of ViT-based methods.** ViT is obviously stronger than Conv. The commonly employed Conv4-64 backbone comprises 113K parameters, while ViT-S has 21M parameters. The performance of ViT-based methods surpasses that of the compared ConvNet-based methods by a significant margin. Further performance improvement with ViT backbone is more challenging compared to the ConvNet backbone.
> + **Relatively small-scale module processor.** The commonly employed Conv4-64 backbone comprises 113K parameters, while Compared to the ViT-S backbone, the parameters of module processor are relatively small (444.7K for module processor, 21M for ViT-S), which may result in smaller impact on performance.
>
> **W5: The method needs additional loss term to prevent degenarcy similar to Top-K approaches.**
>
> Yes, we employ the importance loss to prevent degenarcy.
> The importance loss serves to balance the utilization of specialist modules, mitigating the collapse problem that the router selects a few modules repeatedly.
> However, the importance loss defined in Equation 5 is different from the loss in Top-K approach, such as MoE [ref].
> In our work, we define the importance value of $j$-th module $imp(f_j)$ based on the agreement coefficients at the final iteration $\{c_{ij}^{(T)}\}_{i=1}^N$.
> Because the agreement coefficient $c_{ij}^{(T)}$ is accumulated across $T$ iterations, the importance value mixes the routing information across all iterations, which is absent in Top-K approaches.
>
> [ref] Outrageously large neural networks: The sparsely-gated mixture-of-experts layer. ICLR 2017.
>
> **Q2: Can you explain the relationship between y from Equation 4 and the outputs from the modules?**
>
> Sorry for any confusion.
> Equation 4 is designed to compute the cross-entropy loss between model's outputs and labels of sample.
> The variable $y$  corresponds to the label for input sample $x$.
> The variable $o$ represents the logits obtained from the classifier component applied to the aggregated outputs of modules $\{v_j^{(T)}\}_{j=1}^M$.
> This can be expressed as $o=\text{Classifier}(\text{Agg}(\{v_j^{(T)}\}_{j=1}^M))$, where $\text{Agg}$ denotes the aggregation operation, and in our experiments, we utilize the average operation.
>
> **Q5: With only two modules, what is the value of K in the Top-K method in Table 1? And, how does this compare to an Ensemble approach where all modules are activated?**
>
> The Top-K method employs the noisy top-$k$ gating mechanism to select K modules based on the outputs of router.
> With only two modules, we specify the value of K as 1.
> To compare with ensemble approach, we directly adjust the value of K to 2 and remove the noisy gating mechanism.
> Notably, We find that despite the ensemble variant achieves similar test accuracy on the seen digital compositions (approximately 96\%), its performance decreases for the unseen digital compositions (approximately 43\%).
> We attribute this discrepancy to the absence of modular inductive bias for ensemble approach, preventing it from effectively selecting and composing modules to deal with unseen digital compositions.
>
> **Q6: Could you provide more insight on what occurs in every iteration of the proposed method, and elaborate on the notion of agreement in this context?**
>
> We provide a figure (see Fig 8 in Appendix D) to explain what occurs in every iteration.
> At the initial stage (t=0), both modules $f_1$ and $f_2$ are activated in response to the input image.
> As the iteration proceeds, module $f_1$ gradually deactivates, while module $f_2$ remains unchange.
> In the iterative dynamics, the input-output agreement can be treated as if it was a loglikelihood and is added to the previous logit $a_{ij}^{(t-1)}$, as expressed in Line 3 of Algorithm 1.
> Guided by the accumulated loglikelihood $a_{ij}^{(t)}$, the router models a conditional distribution to **classify** which specailist modules can acquire the input vector $s_i$ (as expressed in Line 4 of Algorithm 1), although there does not exist ground-truth labels for the classification task.

---

> ### Author Response · Authors · 2023-11-20
> **Rebuttal by Authors (3/3)**
>
> **Q8: Is there a plan to conduct experiments showcasing the scalability of the method, particularly when adapting to a new task by adding new modules without forgetting old tasks?**
>
> Thanks to this valuable suggestion.
> We construct a simple continual learning scenario where the min-max digital task and parity code task come sequentially.
> In this scenario, the first task comprises sufficient training data, while the second task only contains limited training data.
> Solving this problem with SMN is straightforward by applying different specialist modules based on the task ID, as parameter-isolation methods in task incremental learning [ref].
> However, we explore a more difficult setting to demonstrate the scalabity of agreement router, where the task ID is unavailable in the module selection process.
> More specifically, We train SMN with 2 modules on the first task, and then adapt it to the second task by adding 1-2 new modules.
> During test, the agreement router cannot acquire the task information to select specialist modules.
> The results are as follow:
>
> |      |  min-max | parity code |
> |-------|-------------|----------|
> | add=0 |  96.12 (0.13)    | 87.48 (3.98)       |
> | add=1 |  95.42 (0.35)    | 88.61 (3.31)       |
> | add=2 |  94.50 (1.73)    | 89.32 (2.34)       |
>
> We find that introducing new modules for the parity code can improve the performance, meanwhile only causing a little performance decrease in the min-max task.
> We attribute this to the scalablity of agreement router, as we observe that agreement router adaptively deactivate the newly added modules for the old task.
> We also find that fine-tuning $W_a$ on the new task will hurt the performance in the old task, as the learned routing information on old task loses.
>
> | Fine-tune $W_a$     |  min-max | parity code |
> |-------|-------------|----------|
> | add=0 |  78.17 (5.27)     | 91.5 (3.8)       |
> | add=1 |  75.19 (6.47)     | 92.2 (3.3)       |
> | add=2 |  73.57 (5.51)     | 92.6 (2.2)       |
>
> [ref] Progressive Neural Networks. Arxiv 2016.

---

> > ### Comment · Reviewer_nZgT · 2023-11-21
> >
> > Thanks for your clarifications, and I wish you all the best!

---

### Author Response · Authors · 2023-11-20
**Global Response**

We are grateful to all reviewers for the effort they have made in reviewing our paper and the valuable comments. We highlight a few enhancements we have made during the rebuttal period.
Detailed revision and additional experiments can be found in the updated version (highlighted by red color).

**Computation Cost.**
+ Reviewer nZgT, 9n4d and y9sx ask the computational cost analysis of SMN to understanding the trade-off between computational burden and accuracy.
We response with a table (in Appendix C.4) to report the FLOPs, training and inference time (seconds), and accuracy for our SMN and other modular variants in the min-max game environment.
We can find that with comparable training and inference time, SMN outperforms other methods in term of accuracy.
Increasing iteration number $T$ can improve the performance of SMN further, while bringing more computational burden.

**Generality of SMN.**
+ **Against Catastrophic forgetting.** Reviewer nZgT and Zqvo concern about the forgetting problem of SMN when adapting to a new task by adding new modules.
We response with a simple experiment in the continual learning scenario (in Appendix E.2) to demonstrate that agreement router can adaptively deactivate the newly added modules for the old task, mitigating catastrophic forgetting and enabling better generalization ability.

+ **Diverse tasks and datasets.**
Reviewer 9n4d, y9sx and Zqvo request additional experiments on a wider range of datasets and tasks.
We response with the results of SMN on the continual learning, reinforcement learning and cross-domain few-shot learning tasks (in Appendix E).
These results demonstrate the effectiveness of our SMN and the agreement router in diverse tasks and datasets.

**Ablation Study.**
+ **Impact of adaptation parameters.** Reviewer nZgT requests some experiments on adaptation parameters, like $W_a$ and $W_c$. We response with ablation study for these parameters (in Appendix C.3) to explain their impact on performance.

+ **Explanation for the iterative dynamics.** Reviewer nZgT requests more explanation about the iterative process in agreement routing. We response with a figure (in Appendix D) to illusrate how different modules changes when iteration proceeds, and explain the input-output agreement as the log-likelihood for allocating input vectors to specific specialits modules.

+ **Multiple layers of modules.** Reviewer nZgT asks how the SMN works with multiple layers of modules. We response with an experiment by stacking multiple layers of modules in the module processor. We find that stacking more layers can further improves the performance of SMN.

---

### Meta-Review · Area_Chair_hxg9 · 2023-12-17

**Metareview:**

The proposed scalable modular framework learns routing among a set of modules, which demonstrates the adaptability and effectiveness to a new task after training on a different task. Reviewers mentioned the computation cost, the generality of SMN, explanation for the iterative dynamics, and other ablations. Authors give detailed feedbacks and satisfied all reviewers. The ACs thus decided to accept it.

**Justification For Why Not Higher Score:**

The routing idea have been used in other field.

**Justification For Why Not Lower Score:**

The SMN  represents a significant advancement in the realm of adaptive learning.

---

### Decision · Program_Chairs · 2024-01-16

Accept (poster)